# Exceptional energy harvesting from coupled bound states

Felix Kronowetter [1,2,3] ✉, Anton Melnikov[4], Marcus Maeder [1], Tao Yang [1], Yan Kei Chiang [2], Sebastian Oberst [3], David A. Powell [2] & Steffen Marburg[1]

Sustainable and affordable energy is one of the most critical issues facing society. Noise is ubiquitous, albeit with a low energy density, making it an almost perfect energy source. Bound states in the continuum overcome this problem through a highly localized energy increase. Here, we present theoretical, numerical, and experimental studies on bound state acoustic harvesters. Under white noise excitation, the bound state harvester outperforms the conventional Helmholtz resonator harvester by a factor of 2.2 in terms of amplitude spectral density of the output voltage and by a factor of 10 in terms of output power. A super-bound state is formed by using pressure coupling in a pseudo-free field environment, further increasing the energy enhancement. This results in a 50-fold increase in output voltage compared to a single bound state harvester. Our findings advance the state-of-the-art in sustainable energy harvesting for low-power devices.

As fossil fuels become increasingly scarce and preventing environmental degradation becomes a priority, finding sustainable ways to generate energy is a cornerstone of the global strategy to mitigate climate change. In 2015, governments worldwide committed to the Paris Agreement and agreed on reaching 17 United Nations Sustainable Development Goals (SDGs, https://sdgs.un.org/goals) with 169 targets by 2030. Especially, SDG 13 relates to climate action with its primary goal being to take urgent action in combating climate change and its impacts (https://sdgs.un.org/goals/goal13). The SDG-compatible energy demand increases in developing regions. While important gaps remain, for many of the SDGs they can be closed by 2050, consistent with the 1.5 °C target and several other planetary boundaries[1].

Yet, little attention is paid to the fact that climate change, particularly extreme weather events, disproportionately affects poorer and minority communities[2]. As a result, the need for low-cost, low-maintenance, sustainable power generation equipment has never been greater. Promising research breakthroughs are easing the challenges of transitioning to sustainable energy sources and ensuring energy security for future generations. Recent advances in solar technology[3–5],

wind power[6,7], and hydropower[8–11] are promising developments in sustainable energy harvesting.

Moreover, with the rapid development of artificial intelligence in recent years, it can further contribute to achieving the goals of the SDGs[12,13]. Innovative ideas such as using piezoelectric microelectromechanical systems to harvest wind energy[14], mimicking energy harvesting processes in nature[15], developing piezoelectric nanogenerators[16] as well as triboelectric nanogenerators[17–19], and turning public transportation depots into profitable energy hubs are part of the latest research[20].

One of the growing trends is harnessing environmental noise for energy generation. Acoustic energy harvesting converts sound energy into electrical energy[21–23]. Moreover, acoustic energy is an indispensable free energy source. It is often disruptive, as we live in a noisy world. However, unlike other sustainable energy sources, sound waves have a low energy density[23], which has not yet made them a significant alternative in large-scale power generation. Thus, based on current knowledge, acoustic energy harvesting will hardly be used to replace currently available energy sources such as fossil fuels or wind and solar energy. However, it has great potential on a smaller scale. For example,

[1]Chair of Vibro-Acoustics of Vehicles and Machines, Department of Engineering Physics and Computation, TUM School of Engineering and Design, Technical University of Munich, Bavaria, Germany. [2]School of Engineering and Information Technology, University of New South Wales, Canberra, ACT, Australia. [3]School of Mechanical and Mechatronic Engineering, Centre for Audio, Acoustics and Vibration, Faculty of Engineering and IT, University of Technology Sydney, Sydney, ACT, Australia. [4]Bosch Sensortec, Dresden, Germany. ✉e-mail: felix.kronowetter@tum.de

it has applications for powering low-power devices[24–26], such as medical implants and sensors.

Recent advancements have focused on various mechanisms and materials to enhance the efficiency and practicality of acoustic energy harvesting systems. For instance, piezoelectric and flexoelectric materials, often integrated with phononic crystals and metamaterials, have significantly improved energy conversion efficiency[27–29]. Devices incorporating these materials can harvest energy from low-frequency sounds, which are prevalent in urban environments and typically difficult to absorb or isolate using conventional methods. Due to the low energy density of acoustic waves, acoustic energy enhancement measures are inevitable. Thus, the use of Helmholtz resonators[30,31] and straight tube resonators[32] has been explored. Moreover, acoustic metamaterials, with their ability to manipulate and focus sound waves in often unprecedented ways, are a natural fit for these applications[21,33,34]. For trending self-powered acoustic systems, increased energy density is also inevitable[26,35–38]. Energy density amplification of several orders of magnitude has been achieved using the latest energy enhancement devices[39,40]. A further increase in acoustic energy density is needed to make low-density acoustic waves a widely usable energy source, though.

Tuning the harvesting systems' quality factors (Q-factors) can enhance energy harvesting efficiency. This strategy allows the use of high-Q resonances. Bound states in the continuum (BICs) are modes with a theoretically infinite Q-factor. Neumann and Wigner established them in a quantum system[41]. Since BICs rely on wave functions, the concept applies to acoustics[42]. Thus, BICs are also present in open acoustic systems and have been the subject of intense research in recent years[43–55]. Under realistic conditions, thermo-viscous losses limit the Q-factor in the acoustic system. In a previous article, the authors investigated maximizing the pressure increase in a resonant cavity that houses a Friedrich-Wintgen (F-W) BIC[49]. These findings, applied to acoustic energy harvesting, form the basis of this article. Extreme Q-factors and unparalleled pressure build-up, resulting in higher energy density, are ideal for acoustic energy harvesting.

Regarding the characterization of acoustic energy harvesters, recent work has focused mainly on the following physical quantities: The system's open circuit voltage[56–58], its output power or output power density[26,59], and its energy conversion efficiency[37,56,60,61]. If no load is connected to the acoustic energy harvester, the open circuit voltage is the highest voltage that the harvester can generate. As the open circuit voltage is related to the pressure increase of the resonant structure and the properties of the piezoelectric transducer, it is a suitable quantity to characterize pressure-enhancing structures.

Here, we introduce BIC-based acoustic energy harvesting as a technically viable mechanism to address some of the energy requirements of future applications. Theoretical and numerical investigations provide fundamental insights into the formation mechanism of the F-W BIC and the resonant cavity in which it is housed. Furthermore, we analyze the BIC harvester numerically and experimentally regarding output voltage and compare it with a state-of-the-art Helmholtz resonator harvester. Impedance tube measurements and laser Doppler vibrometry help to verify the results. We compare the two harvesting systems on the basis of open-circuit voltage. We then present optimum resistance studies for both harvesters and compare their measured output power. Finally, we present a coupling-enhanced BIC harvester, with appropriate experiments to demonstrate its performance.

## Results

A F-W BIC is constructed in an open system to enhance the acoustic energy harvesting efficiency. F-W BICs are based on the destructive interference of degenerate modes of the same symmetry. They are entirely embedded in the continuum near the point of modal degeneracy of the closed system, and they still form even if the system's symmetry is broken[62]. In summary, F-W BICs are predictable using the closed system solution and are stable regarding asymmetries in the system[49].

### Bound state in the continuum (BIC) prediction

The Hermiticity of a Hamiltonian is one of the key postulates of quantum mechanics, which requires that a closed system has real eigenenergies $E$. However, an open system exchanges energy between a subsystem and its environment. Open systems are a prerequisite for the formation of F-W BICs. Theoretically, an effective non-Hermitian Hamiltonian $\boldsymbol{H}$ describes open systems. The number of modes, which determines the size of the Hamiltonian, can be reduced to a reasonable number[63]. Since the F-W BIC analyzed in this article is based on destructive interference of two degenerate modes, the two-mode approximation leads to a two-level effective non-Hermitian Hamiltonian

$$\boldsymbol{H} = \begin{bmatrix} E_{m,n} & 0 \\ 0 & E_{n,m} \end{bmatrix} - \mathrm{i}k_p \begin{bmatrix} W_{m,n}^2 & W_{m,n}W_{n,m} \\ W_{n,m}W_{m,n} & W_{n,m}^2 \end{bmatrix}. \tag{1}$$

Here, $W_{m,n}$ denotes the coupling constants, where $m$ and $n$ are the modal indices that indicate the pressure maxima along the corresponding axes. See Supplementary Information, Section S1 for the detailed derivation of the analytical formulations. Furthermore, the assumed uniform pressure distribution in the z-direction leads to the two-dimensional simplification. In addition, $k_p$ denotes the wavenumber of the channels that couple the closed system to the continuum, which depends on the modal index $p$ of its transverse modes. Furthermore, we assume that the coupling to the evanescent channels ($p > 1$) can be neglected in this case since their coupling constants are insignificantly small[64]. In this case, the F-W BIC results from a particular configuration of the geometric parameters of a resonant cavity attached to a waveguide. The cavity is a triangular volume attached to a tube, henceforth referred to as a fully reduced cavity (FRC). A two-dimensional schematic of the geometry provides Fig. 1a.

The dimensions of the resonant cavity are chosen so that an excessive pressure peak and maximum output voltage can be observed at identical frequencies regarding the numerical data. The modal field of the numerically simulated quasi-BIC (QBIC) in terms of absolute pressure values, denoted by the mode M₃₃₁, is shown in Fig. 1b. Following Lyapina et al.[64], $M_{pqr}$ denotes the leaky modes, where $p$, $q$, and $r$ are the number of maxima in the pressure field along the x-, y-, and z-axis, respectively. The position of the pressure peak is at the lower edge of the FRC, i.e., at the position of the origin of the coordinate system in Fig. 1a. A piezoelectric transducer (PZT) of type ARD VIB 01 is attached to the cavity at this point to maximize energy harvesting. The thick blue line in Fig. 1a indicates the PZT. Figure 1c shows the 3D printed three-dimensional sample for impedance tube measurements. Furthermore, we numerically determine the exact geometry using parameter sweeps that map the maximum sound pressure and the output voltage to the frequency and the cavity length. See Supplementary Information, Section S2, Supplementary Fig. S3 for the corresponding mapping plots. The cavity length of the sample is selected to match the maximum output voltage of the mapping. Thus, the cavity length is set to $L_x = 83.5$ mm, its height to $L_x = 80$ mm, and its thickness to $L_x = d = 40$ mm, equal to the diameter of the tube. Figure 1d shows the numerical (orange dashed line) and the measured (blue solid line) absorption coefficients of the FRC. They agree with maximum absorption at $f = 2074$ Hz, with the measured one showing higher attenuation. We explain this behavior by the influence of additional structural damping, which is not represented by the numerical data using sound hard boundary conditions. Figure 1e shows the real parts of the eigenvalues of the system for the full destructive interference pattern of modes M₃₁₁ and M₁₃₁. The thin, solid black lines show the coupled mode theory results for a closed cavity. As a result, the intersection of the black lines is the point of modal

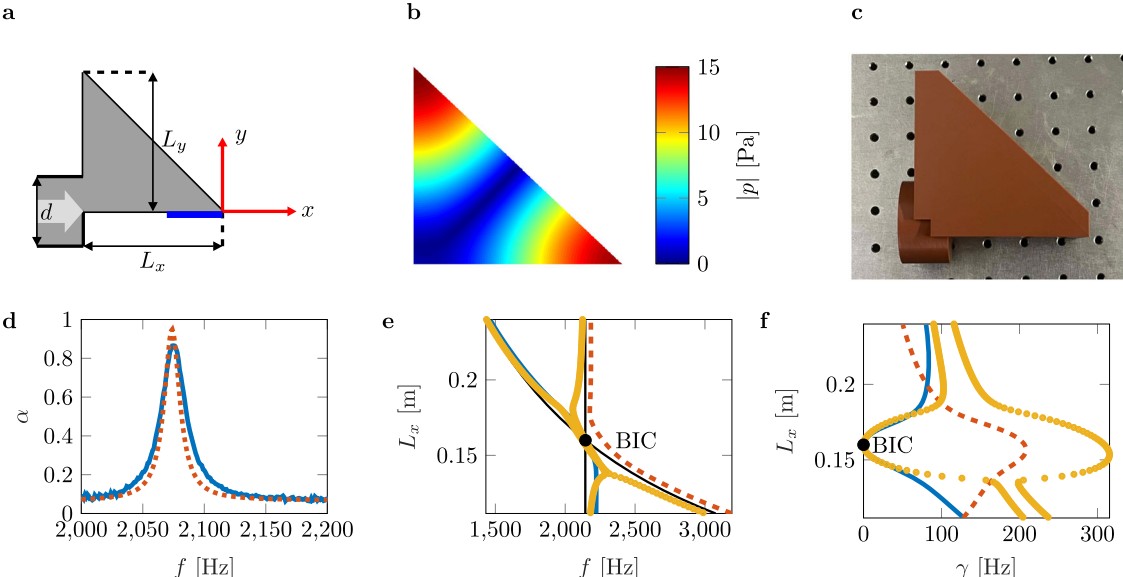

**Fig. 1 | Bound state in the continuum (BIC) configuration. a** Schematic of the fully reduced cavity (FRC) coupled to an acoustic waveguide. The $z$-axis is perpendicular to the $xy$-plane. **b** Numerically simulated BIC mode ($M_{331}$) in terms of absolute pressure values. **c** 3D printed sample of the FRC. The length is set to $L_x = 83.5$ mm, the height to $L_y = 80$ mm, and the expansion in $z$-direction is equal to the diameter of the tube $L_z = d = 40$ mm to ensure a two-dimensional wave pattern. **d** Absorption coefficients of the numerical data (dashed orange line) and the impedance tube measurements (solid blue line). **e** Avoided crossing of the real parts of high-Q and low-Q modes for varying length $L_x$ of the resonant cavity. **f** Vanishing imaginary part of the BIC mode.

degeneracy. It is an accurate predictor of the BIC. The solid blue and dashed orange lines are the results of the numerical model. The yellow dots denote the real parts of the eigenvalues of the two-level non-Hermitian Hamiltonian for the modes $M_{311}$ and $M_{131}$. All three models accurately predict the BIC. The imaginary parts of the eigenvalues obtained by numerical simulation (blue and orange lines) and by solving the Hamiltonian (yellow dots) appear in Fig. 1f. For the imaginary parts, the Hamiltonian accurately predicts the BIC but over-predicts the values of the highly damped mode due to considering only two modes.

**Experimental verification of maximum output voltage**

The response of the harvesting system is related to both the acoustic pressure within the resonant structure and the characteristics of the PZT. The impedance of the circuit, and therefore, amongst others, its resistance, determines the system response in terms of output power. To eliminate the effect of resistance on the output signal, we use the open circuit voltage. As the output voltage is linearly correlated with the pressure enhancement, we use it as the physical quantity to characterize and compare the systems presented. Three configurations are manufactured and compared with each other. The first configuration is a 3D printed sound hard termination with mounted PZT to determine the resonance of the mounted PZT. The second configuration is a harvester based on a Helmholtz resonator to demonstrate the harvesting properties of the system. For a graphical representation of the first and second configurations, see Supplementary Information, Section S3, Supplementary Fig. S4a. The third configuration is a 3D printed three-dimensional sample of the FRC with the PZT mounted on it (Fig. 2a).

For all three samples, we measure the absorption in an impedance tube. The Supplementary Information, Section S3, Supplementary Fig. S4b provides the results of the sound hard termination and the Helmholtz resonator. The sound hard termination shows an absorption peak $\alpha = 0.36$ at $f = 2307$ Hz. This is identified as the resonance of the mounted PZT, which leads to dissipation in the system. The Helmholtz resonator has a maximum absorption of $\alpha = 0.64$ at $f = 2036$ Hz. The FRC has a maximum absorption of $\alpha = 0.95$ at $f = 2074$ Hz

(Fig. 1d). In addition, a Laser Doppler Vibrometer (LDV) is focused on the PZT to measure its modal field while it is acoustically excited, see Fig. 2b. This is necessary to ensure the correct mounting and alignment of the PZT (Supplementary Information, Section S3, Supplementary Fig. S5a, b).

As a reference, both, the PZT and Helmholtz resonator harvester are placed in an impedance tube. The system is excited using white noise in a frequency range of $f = 2000$ Hz to $f = 2400$ Hz at 100 dB(A). We measure the PZT velocity's amplitude spectral density (ASD) $\tilde{v}$ in mm/s/$\sqrt{\text{Hz}}$ and the peak output voltage's ASD $\tilde{U}$ in mV/$\sqrt{\text{Hz}}$ of the PZT in parallel using the data acquisition system of the LDV (Fig. 3a).

The ASDs of the velocities and output voltages of the PZT show local maxima with $\tilde{v} = 0.042$ mm/s/$\sqrt{\text{Hz}}$ at $f = 2311$ Hz and with $U = 0.09$ mV/$\sqrt{\text{Hz}}$ at $f = 2302$ Hz. The Helmholtz resonator harvester also shows a local maximum for the ASD of the velocity $\tilde{v} = 0.042$ mm/s/$\sqrt{\text{Hz}}$ and the ASD of the output voltage $\tilde{U} = 0.09$ mV/$\sqrt{\text{Hz}}$ at $f = 2355$ Hz and $f = 2360$ Hz, respectively. The resonance of the PZT compared to the one mounted on the Helmholtz resonator is thus shifted by about 50 Hz. The Helmholtz resonator harvester also shows a second local maximum corresponding to the acoustic resonance at 2045 Hz. However, an important consideration for the BIC is the acoustic resonance, where the ASD of the velocity is 0.003 mm/s/$\sqrt{\text{Hz}}$ higher, hence $\tilde{v} = 0.045$ mm/s/$\sqrt{\text{Hz}}$, than for the PZT resonance. The ASD of the output voltage of the acoustic resonances has its maximum value at $\tilde{U} = 0.118$ mV/$\sqrt{\text{Hz}}$.

After establishing the reference, we measure the FRC (Fig. 3b). The ASD values of the velocity and the output voltage maximize at the same frequency $f = 2068.5$ Hz, which means that there is a slight shift of $f = 5.5$ Hz compared to the maximum absorption at $f = 2074$ Hz. For a white noise excitation, we obtain a maximum ASD value of the velocity of $\tilde{v} = 0.16$ mm/s/$\sqrt{\text{Hz}}$ and a maximum ASD value of the output voltage of $\tilde{U} = 0.25$ mV/$\sqrt{\text{Hz}}$. As expected, the second local maximum appears at 2329 Hz with values of $\tilde{v} = 0.046$ mm/s/$\sqrt{\text{Hz}}$ and $\tilde{U} = 0.063$ mV/$\sqrt{\text{Hz}}$, similar to the ones of the PZT resonance observed before.

Furthermore, we excite the system with the determined target frequency using a sinusoidal signal at $f = 2068.5$ Hz. The corresponding

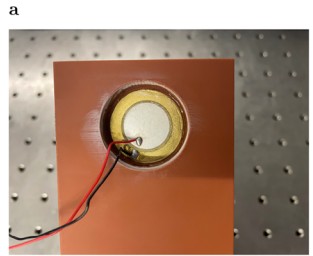
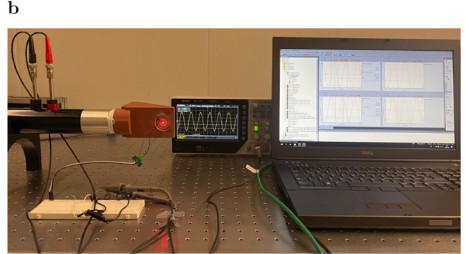

**Fig. 2 | Experimental setup. a** Printed sample of the fully reduced cavity (FRC) with mounted piezoelectric transducer (PZT). **b** Experimental setup for impedance tube and Laser Doppler Vibrometer (LDV) measurements.

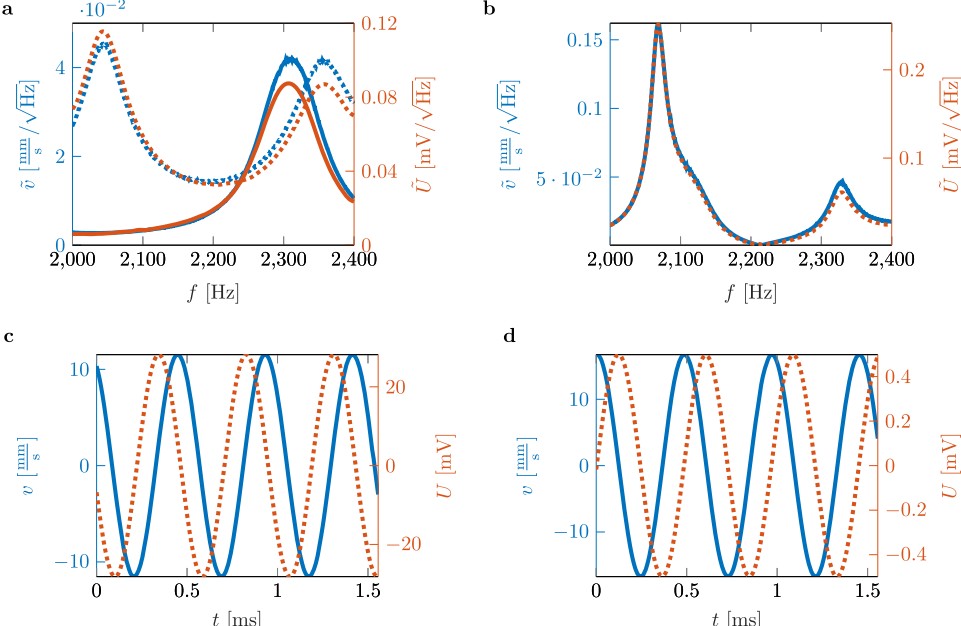

**Fig. 3 | Amplitude spectral densities (ASDs) of the velocity and output voltage of the piezoelectric transducer (PZT). a** Sound hard termination with PZT (solid blue and orange lines) and Helmholtz resonator with PZT (dashed blue and orange lines) under white noise excitation. **b** Fully reduced cavity (FRC) with PZT under white noise excitation. **c** Velocity and output voltage of the system excited at the previously determined target frequency of $f = 2068.5$ Hz. **d** Velocity and output voltage of the system excited at the previously determined target frequency of $f = 2068.5$ Hz with a 10 $\Omega$ resistor embedded in the circuit.

velocity and output voltage provides Fig. 3c for 1.5 ms. Throughout all measurements, the LDV captures the structural dynamics of the PZT. Numerical data indicate that, compared to a Helmholtz resonator peaking at the same resonant frequency, the output voltage is 192 times higher.

### Experimental verification of maximum output power

The output power of the FRC harvesting system is studied numerically to analyze its harvesting characteristics. Therefore, an optimal resistance study is performed. The results are shown in Supplementary Information, Section S5, Supplementary Fig. S7. Figure 3d shows the harvesting characteristics of the FRC system. The system is again excited at $f = 2068.5$ Hz, but with a 10 $\Omega$ resistor placed in the circuit. The voltage is measured across the resistor to determine the electrical current. The velocity of the PZT increases to 16.7 mm/s as the voltage drops to 0.5 mV. In this way, an electric current of 50 μA is measured. More importantly, we can calculate the electrical power extracted from the circuit as $P = U_{rms}^2/R = (U/\sqrt{2})^2/R = 12.5$ nW. Measurements of three additional resistors, 100 $\Omega$, 1 k$\Omega$, and 10 k$\Omega$, are provided for better evaluation of the system. This results in the following output powers of 121 nW, 822 nW, and 479 nW respectively. The maximum output power of 822 nW with a 1 k$\Omega$ resistor placed in the circuit is in line with the trend of the numerical prediction. The system's energy

conversion efficiency is therefore around 10%. Furthermore, the sound hard termination with mounted PZT has a maximum output power of 50 zW and the Helmholtz resonator of 87 nW. In summary, the FRC outperforms the Helmholtz resonator by a factor of about 10 in terms of output power.

### Coupled bound states in the continuum (BICs) for acoustic energy harvesting

Up to this point, we outline the performance of a BIC harvester. To the best of our knowledge, all studies on acoustic BICs have been carried out in a confined environment, i.e., a geometry placed in a waveguide. For practical reasons, an acoustic harvester should operate under realistic conditions. Therefore, we present a setup that demonstrates the harvesting characteristics of BIC harvesters in a pseudo-free field environment. Here, we understand a pseudo-free field as a configuration where the sample is placed at least one wavelength away from the sound source but with reflective outer boundaries in an attempt to simulate realistic conditions. With the open impedance tube as the sound source, the sample is placed at a distance three times the resonance wavelength to mimic far field conditions. The sound source and sample are positioned in an even plane, where we measure three different configurations. Figure 4 visualizes the setup and the samples used.

A graphic representation of the introduced FRC provides Fig. 4a. In the second configuration, two FRCs are placed side by side on the centerline of the impedance tube (Fig. 4b), which allows us to see if two FRCs in close proximity are interfering. This vital insight helps to design FRC arrays for future applications. The third configuration (Fig. 4c), consists of two FRCs in a rectangular housing. The fundamental idea, based on previous work by the authors[50,65], is to further increase the pressure enhancement based on BIC coupling. Therefore, two FRCs are coupled by means of a rectangular geometry (Fig. 5a).

One FRC has a constant cavity length. The second has a variable cavity length. The system is excited at 1 Pa. As two FRCs with different cavity lengths couple via the distance between them, the frequency-dependent coupling significantly influences the overall system behavior[65]. A secondary effect but the most essential property of coupling two BICs in unbounded domains is the formation of another BIC. See the Supplementary Information, Section S4, Supplementary Fig. S6 for more details. At the resonant frequency of the FRC with constant cavity length, a main peak and several other Fano resonances with variable Q-factors are observed throughout the parameter plane. The Fano peaks disappear in a particular configuration, and another BIC is discovered. This occurrence is indicated by the white cross in Supplementary Information, Section S4, Supplementary Fig. S6.

The black cross marks the maximum pressure enhancement. We use this information to simulate a finer mapping around that point, see Fig. 5b for the corresponding results. With these results, we adjust the cavity length and the distance between the two cavities to determine a specific configuration. Here, the cavity length of the first cavity is set to $L_{x,1}$ = 83.5 mm and the length of the second one to $L_{x,2}$ = 81 mm. The distance between the cavities is set to $\Delta$ = 73 mm. The black crosses mark the two corresponding Fano peaks associated with maximum

pressure enhancement. The sound pressure inside the two FRCs is significantly increased by a factor of about 40 and measured Q-factors of 255 and 162, respectively. Thus, this BIC formation results in a super-BIC with exceptional pressure enhancement capabilities.

Here, we measure all three configurations in the pseudo-free field setup. The open-ended impedance tube is used as the sound source for all measurements, emitting a white noise excitation with a sound pressure level of 70 dB measured at the tube microphones to mimic a realistic excitation (We consider staying below the upper limit for safe operation in applications (e.g. washing machine noise levels, etc.)).

The first step is to measure the FRC. The solid blue line in Fig. 6a represents the ASD of the output voltage. The dominant peak is observed at $f$ = 2068.5 Hz with $\tilde{U}$ = 0.05 mV/$\sqrt{\text{Hz}}$. Compared to the impedance tube measurement results shown in Fig. 3b, the ASD of the output voltage drops by a factor of 50. The immense voltage drop illustrates the significant difference between an ideal impedance tube setup in a laboratory environment and real-world conditions.

The second step is to place a second FRC next to the first one and center the configuration, i.e., see Fig. 4b. The purpose of this configuration is to demonstrate an arrangement and its effect on the performance of each FRC. The result is shown in Fig. 6a by the dashed orange line. It matches the result of the single cavity in terms of ASD of the output voltage. Hence, the ASD of the output voltage is constant using two FRCs.

The third step is to measure the harvesting capabilities of the super-BIC. The numerical results show an extraordinary pressure enhancement for each cavity, i.e., see the black crosses in Fig. 5b. The ASD of the output voltage for each cavity is measured and plotted in Fig. 6b. The solid blue line marks the ASD of the output voltage for the second cavity. The dominant peak is observed at $f$ = 2088.5 Hz with

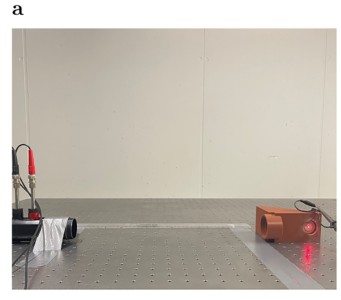
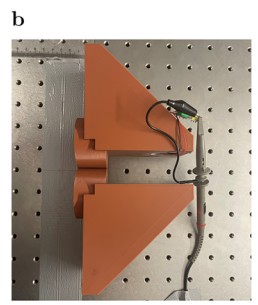
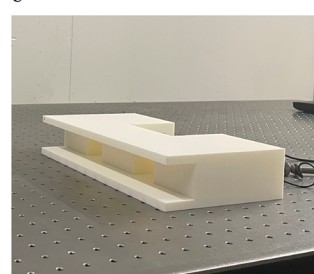

**Fig. 4 | Pseudo-free field measurement setup. a** Measurement setup with a single fully reduced cavity (FRC). **b** Measurement setup with two FRCs placed at the center line. **c** Measuring setup with two FRCs coupled in a rectangular housing.

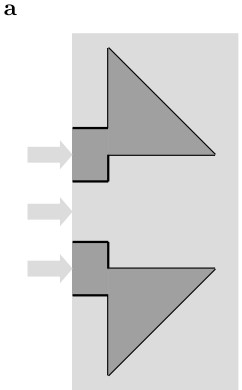
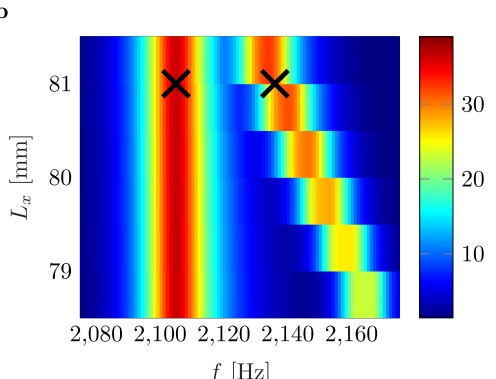

**Fig. 5 | Coupled bound states in the continuum (BICs). a** Schematic of two fully reduced cavities (FRCs) embedded in a rectangular housing. **b** Mapping of the maximum pressure enhancement of the coupled FRCs in terms of frequency, cavity length, and distance ranging from $f$ = 2075 Hz to $f$ = 2175 Hz, $L_{x,2}$ = 78.5 mm to $L_{x,2}$ = 81.5 mm, and $\Delta$ = 68 mm to $\Delta$ = 76 mm, respectively. The black crosses mark the two Fano peaks associated with maximum pressure enhancement.

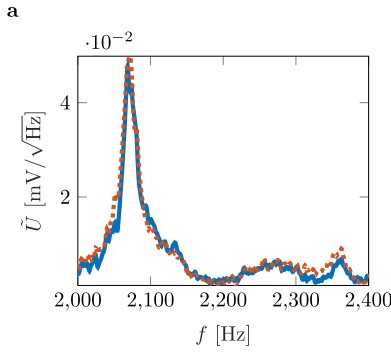
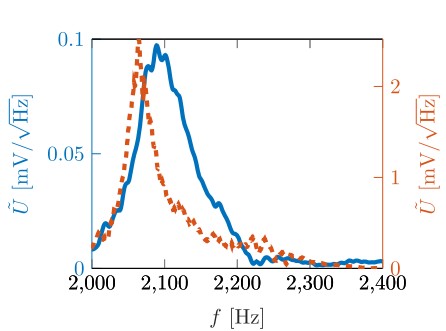
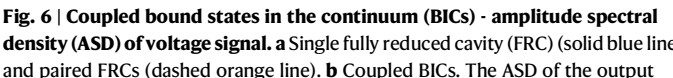

**Fig. 6 | Coupled bound states in the continuum (BICs) - amplitude spectral density (ASD) of voltage signal. a** Single fully reduced cavity (FRC) (solid blue line) and paired FRCs (dashed orange line). **b** Coupled BICs. The ASD of the output

voltage of the first cavity is marked by the dashed orange line. The ASD of the output voltage of the second cavity is marked by the solid blue line.

$\tilde{U} = 0.1\,\mathrm{mV}/\sqrt{\mathrm{Hz}}$. The ASD of the output voltage is doubled compared to the single FRC and double FRC. The ASD of the output voltage for the first cavity is even more striking, marked by the dashed orange line. The dominant peak is observed at $f = 2064.5\,\mathrm{Hz}$ with $\tilde{U} = 2.5\,\mathrm{mV}/\sqrt{\mathrm{Hz}}$, which means that a factor of 50 increases the ASD of the output voltage compared to the single FRC and the paired FRCs. As a result, the Super BIC provides an extraordinary increase in pressure and significantly improved harvesting capabilities.

## Discussion

We report on the theoretical prediction of a BIC and the numerical design up to the experimental verification of BIC-based acoustic energy harvesting. As for the theoretical model, the accuracy of the two-level Hamiltonian is limited. One possible solution is to include coupling to the evanescent channels. The model can be further improved by including more modes and moving away from the two-level approximation[48]. The accuracy of the current models is sufficient in our case.

Our studies show that the presented BIC harvester is superior to a conventional Helmholtz resonator harvester by a factor of 2.2 in terms of ASD of the output voltage. The BIC results in an increase in stored energy. Adding a PZT with a resistive load results in a more efficient coupling of energy into the PZT. Thus, we are able to achieve such an enhancement thanks to the improved impedance matching between the medium and the PZT[66]. Furthermore, we demonstrate the harvesting performance of the proposed BIC harvester by connecting a resistor to the terminals of the PZT to simulate an electrical load.

We also introduce the concept of coupled BICs that form a super-BIC with exceptional energy enhancement properties. We present what we believe to be the first non-waveguide measurements of BICs, demonstrating the harvesting capabilities of the super-BIC under white noise excitation. Compared to a single BIC, the super-BIC can increase output voltage by a factor of 50. The output voltage could be further increased using more efficient PZTs or nanogenerators[16–19].

BICs are ideal for acoustic energy harvesting. However, the high-Q Fano resonances responsible for the required energy enhancement have a major drawback: they are very narrowband[67]. Compared to state-of-the-art broadband acoustic energy harvesting combining nonlinear magnetic force with sonic crystal metamaterials[68], using piezoelectric polyvinylidene fluoride harvesters[69], and acoustic black hole harvesters[61], which can cover frequency ranges from 50 to 800 Hz, BIC harvesters are far more sensitive and efficient at converting acoustic energy into electrical energy due to better impedance matching. Since the Helmholtz equation is linearly scalable, we can get around the narrowband problem by creating an array of BICs with variable Fano resonances. Figure 5b suggests that for a given choice of

parameters, two Fano peaks can converge to cover a wider frequency range. This enables broadband BIC-based harvesting.

The global growth momentum for wind and solar energy lags behind the growth needed to meet global climate goals regarding the 1.5 °C target, even when the technology is available[70]. The socio-economic benefits of sustainable energy are that it does not just address climate change. It also promotes public health and economic growth[71,72]. Recognizing that climate change disproportionately affects poorer communities[2], we are demonstrating a sustainable, low-cost, do-it-yourself BIC harvester. We have used off-the-shelf PZTs to make the BIC harvester easy to rebuild. All the materials used are readily available, inexpensive, and sustainable. In addition, the setup presented here can be combined with a simple diode bridge rectifier or with a synchronized switch harvesting on inductor (SSHI) circuit. Our findings are one step towards a greener, cleaner world.

Although the harvester is only suitable for low-power devices, many small steps can make a big difference and help drive the growth needed to meet the 1.5 °C global warming target. Thus, BIC-based acoustic energy harvesting has enormous potential to make low-density acoustic sound accessible for energy harvesting in everyday life.

In conclusion, introducing a BIC harvester will take acoustic energy harvesting to the next level. In particular, the potential for coupled BICs to create a super-BIC for acoustic energy harvesting is enormous. It opens up the following opportunities: off-grid, self-powered sensors, medical implants, and even mobile phone charging when enough energy is generated.

## Methods
### Analytical model
The two-level effective non-Hermitian Hamiltonian for our system reads as

$$\boldsymbol{H}_{eff} = \begin{bmatrix} E_{m,n} & 0 \\ 0 & E_{n,m} \end{bmatrix} - \mathrm{i}2k_1 \begin{bmatrix} W_{m,n}^2 & W_{m,n}W_{n,m} \\ W_{n,m}W_{m,n} & W_{n,m}^2 \end{bmatrix}. \quad (2)$$

For a precise formulation of the terms used and a full theoretical analysis, see the Supplementary Information, Section S1. The mode corresponding to the purely real eigenvalue of the BIC follows as

$$\boldsymbol{X} = \frac{W_{m,n}W_{n,m}}{\sqrt{W_{m,n}^2 + W_{n,m}^2}} \left( \frac{1}{W_{m,n}}, \frac{-1}{W_{n,m}} \right)^{\dagger}. \quad (3)$$

Furthermore, we approximate the eigenfield profile of the Friedrich-Wintgen BIC as

$$\psi_{BIC} \approx \frac{W_{n,m}}{\sqrt{W_{m,n}^2 + W_{n,m}^2}} \psi_{m,n} - \frac{W_{m,n}}{\sqrt{W_{m,n}^2 + W_{n,m}^2}} \psi_{n,m}. \tag{4}$$

## Numerical simulations

All simulations in this article are performed with the commercial software COMSOL Multiphysics (Acoustics Module, Structural Mechanics Module, and AC/DC Module). The speed of sound and the air density is $c_0 = 343$ m/s and $\rho_0 = 1.2$ kg/m³, respectively. We consider the walls of the cavity and the walls of the waveguide to be rigid and thus apply sound hard boundary conditions. We also consider thermo-viscous losses in our system. We apply the no-slip condition for the velocity field and an isothermal condition for the temperature at the walls of the cavity. To ensure non-reflective boundary conditions at the ends of the waveguide, we use perfectly matched layers. An acoustic-structure boundary is applied at the interface between the PZT and the fluid medium. We use strong coupling. This means that the kinetic condition and the kinematic condition at the solid-fluid interface are taken into account. The PZT is modeled using a brass plate facing the fluid medium (density $\rho = 8900$ kg/m³, Young's modulus $E = 100$ GPa, and Poisson's ratio $\nu = 0.3$) with a layer of Lead Zirconate Titanate (PZT-5H: $\rho = 7500$ kg/m³, piezoelectric strain constants $d_{31} = -274$ pC/N and $d_{33} = 593$ pC/N, relative permittivity $\epsilon_{11} = \epsilon_{22} = 1704.4$ and $\epsilon_{33} = 1433.6$) applied to it. In the experimental setup, the PZT is bonded to the printed structure, and assuming the bond is rigid, we apply a fixed constraint to the interface between the structure and the brass plate of the PZT. The Electrostatics interface is used to model the PZT. In addition, an electrical circuit is embedded in the simulation to model the output power of the system in relation to a given resistance.

## Device fabrication

The experimental samples are 3D printed on the Bambu Lab X1-Carbon 3D printer. The material of choice is polylactide (Bambu Lab PLA Basic Filament). All samples are provided with a hole to allow the PZT to be mounted. The PZTs are of type ARD VIB 01. It is a ceramic vibration sensor that emits an analog signal when vibration occurs. The sensor can measure vibrations of different strengths. The PZTs are glued to the designated holes in the samples.

## Measurement

The absorption coefficients of the symmetry-reduced cavities are measured using an AED 1000 - AcoustiTube impedance tube with a diameter of 40 mm. A laser Doppler scanning vibrometer PSV-500 (wavelength of light of 632.8 nm) from Polytec measures the velocity of the excited PZTs using a pseudo white noise excitation on the one hand and a sinusoidal signal on the other hand. For the former, we investigate a frequency range of 800 Hz–3.2 kHz with a 70 dB band-pass filter between 1.8 kHz and 2.6 kHz, a frequency resolution of $\Delta f = 0.5$ Hz and a maximum sensitivity of 5 mm/s. Furthermore, using the pseudo white noise excitation allows the application of a rectangular window for the FFT analysis, which averages ten individual measurements in the complex plane. This spectral analysis helps to identify the peak values for the structural velocity and the corresponding PZT output voltages. For the sinusoidal excitation, utilizing a sampling rate of 160 kHz allows an appropriate signal analysis for the excitation frequency of 2068.5 Hz, corresponding to the peak values of the previous spectral analysis. Utilizing the internal trigger of the data acquisition system facilitates an ensemble averaging of ten individual measurements with a maximum sensitivity of 50 mm/s and a 70 dB bandpass filtering between 1.95 kHz and 2.15 kHz. The peak values of

the velocity are 8 mm/s for the velocimeter and 12.72 mV for the PZT connected to the acquisition system with an input impedance of 1 M$\Omega$. These values correspond to a sound pressure amplitude within the impedance tube of $p_1 = 5.806$ Pa for microphone 1 (black) and $p_2 = 2.662$ Pa for microphone 2 (red), see Fig. 2b.

## Data availability

The data used in this study are available in the figshare database under [https://doi.org/10.6084/m9.figshare.28430336].

## Code availability

The codes used in this study are available in the figshare database under [https://doi.org/10.6084/m9.figshare.28430336]. Additionally, we host a COMSOL server, where we provide free access to vibro-acoustic applications: https://apps.vib.ed.tum.de/app-lib. We particularly created an application to give people an understanding of BICs and their influence on sound attenuation: https://apps.vib.ed.tum.de/app/BIC_TL_EF_App_V02_mph?id=0012.

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

## Acknowledgements

T.Y. acknowledges support from a European Union's Horizon Europe research and innovation program fellowship under Marie Skłodowska-Curie Grant Agreement No. 101063867.

## Author contributions

F.K., A.M., M.M., S.O., D.A.P., and S.M. conceived the project. F.K. derived and implemented the theoretical model. F.K. designed the geometry and modeled the physics behind it. F.K., M.M., and T.Y. made samples and performed measurements to obtain the absorption coefficients, ASDs of the velocity of the PZT, and ASDs of its output voltage. D.A.P., A.M., S.O., and Y.K.C. advised the modeling and experimental process. All authors discussed the results. F.K. prepared the manuscript with contributions from all authors. M.M., A.M., T.Y., S.M., S.O., Y.K.C., and D.A.P. edited the manuscript.

## Funding

## Competing interests

The authors declare no competing interests.
