## [Transparent Peer Review file · Nature Communications]

Exceptional energy harvesting from coupled bound states

Corresponding Author: Dr Felix Kronowetter

Version 0:

Reviewer comments:

Reviewer #1

(Remarks to the Author)

Felix Kronowetter et al describe the development of an energy harvesting system that leverages the energy enhancement properties of FW bound states in the continuum, which is achieved by integrating a coupled bound states in the continuum with energy conversion mechanisms such as piezoelectric elements. The authors further explore the application of BIC systems and coupled BIC systems in acoustic energy harvesting. In general, this work presents some interesting results and expands the application range of BICs into energy harvesting. The manuscript can be accepted for publication after addressing the following concerns:

1. In the term of acoustic energy harvesting, the authors primarily use open voltage, especially when comparing the performance of different configurations. The authors shall provide more details to justify why we can use open voltage to describe energy harvesting.
2. In the Introduction, the authors emphasize the importance of acoustic energy harvesting in the field of energy harvesting. To provide a comprehensive overview of this topic, it is recommended to include the latest developments in acoustic energy harvesting, including output power and open voltage. This will help readers better understand the performance characteristics of acoustic energy harvesting and its practical application potential.
3. The authors measure the open-voltage of three different structural configurations to characterize acoustic energy harvesting. However, as voltage does not directly represent energy, using it as an evaluation parameter may not provide an accurate evaluation, energy harvesting should be characterized by parameters such as output power or energy density. Please provide output power of three different configurations.
4. The authors claim that the measured absorption coefficient in Figure 1d is lower than the numerical value due to additional damping introduced by the PZT. It is recommended to provide absorption coefficient data for the structure without the PZT to verify this explanation.
5. The authors measure the output power of the FRC system as 12.5 nW with an external load resistance of 10 Ω . Please clarify the reason for selecting 10 Ω as the external load resistance and indicate whether an optimal resistance calculation was conducted.
6. Please provide the energy conversion efficiency of the system, as this is an important parameter for acoustic energy harvesting systems.
7. The results of the numerical simulation indicate that the output voltage of the FRC structure is 192 times higher than that of the HR resonator. It is recommended to include relevant parameters for output power. Additionally, please explain whether losses were introduced in the simulation and how they were set.
8. In conclusion, the authors claim that they achieve impedance matching between the medium and the PZT, please explain how this is accomplished. As far as I know, the acoustic impedance of the medium and PZT are both significantly higher than that of air, allowing them to be considered as hard boundaries.
9. Since this work focuses on acoustic BICs, some relevant works on this topic shall be cited. Some examples are listed as follows
 - (a) B. Jia et al., Phys. Rev. Appl. 19, 054001 (2023)
 - (b) Z. Zhou et al., Phys. Rev. Lett. 130, 116101 (2023)
 - (c) A. A. Lyapina et al., Journal of Sound Vibration 421, 48 (2018)

Reviewer #2

(Remarks to the Author)

This paper presented theoretical, numerical, and experimental studies on bound state acoustic harvesters. The overall study is very rigorous, but I would like to make the following suggestions here.

1、 Please center the image in Fig.2 and Fig.4.

2、 In this paper, white noise is used for testing. It is well known that the power spectral density of each frequency component of white noise is constant, and the power spectral density of Brown noise is proportional to the reciprocal frequency. So how would the results be different if Brownian noise was used for the test?

Reviewer #3

(Remarks to the Author)

This article presents a novel energy harvester based on BICs. The paper provides a detailed introduction to the design and experimental process. The research findings are innovative. This BICs energy harvester is inspiring for applications. I have the following unclear content.

(1) How does the BICs energy harvester introduced in the paper improve impedance matching between the medium and PZT and enhance acquisition performance? This is not clearly stated in the paper. Suggest adding explanations.

(2) The simulation model used the commercial software COMSOL Multiphysics (Acoustic Module and Structural Mechanics Module). The performance of PZT and BICs is related to the structural connection method and environment. How is this aspect understood and designed? Suggest providing a detailed introduction.

Version 1:

Reviewer comments:

Reviewer #1

(Remarks to the Author)

The authors have thoroughly addressed the reviewers' comments, leading to a substantial improvement in the quality of the paper. Therefore, I recommend its acceptance.

Reviewer #2

(Remarks to the Author)

This article has met the requirements for publication and can be published.

Reviewer #3

(Remarks to the Author)

The author has revised the issue.

A new type of energy harvester based on BICs has been designed. Detailed introduction of the research process and nuclear reality. The research results have implications for applications. There are no suggestions for modification.

Response to the Reviewers

Journal:

Manuscript No.:

NCOMMS-24-70176-T

Title:

Exceptional energy harvesting from coupled bound states

Authors:

Felix Kronowetter, Anton Melnikov, Marcus Maeder, Tao Yang, Yan Kei Chiang, Sebastian Oberst, David A. Powell, and Steffen Marburg

General Remarks

Dear Reviewers:

We would like to express our appreciation for the efforts made by the reviewers to improve our paper. The comments are very constructive and helpful in improving the quality and presentation of the paper. We have done a substantial amount of work to incorporate the suggestions and address the concerns raised by the reviewers. In response to reviewer #1's request, we have performed numerical simulations and put considerable effort into new measurements. In response to reviewer #2, we have discussed all concerns. In response to the requests of reviewer #3, we have added more data and more detailed explanations to the manuscript to highlight the novelty, innovation, and scientific impact of our work. **Red-colored text indicates changes made to the manuscript**, while **blue-colored text highlights the authors' responses**.

Specific Response to Reviewer #1:

Felix Kronowetter et al describe the development of an energy harvesting system that leverages the energy enhancement properties of FW bound states in the continuum, which is achieved by integrating a coupled bound states in the continuum with energy conversion mechanisms such as piezoelectric elements. The authors further explore the application of BIC systems and coupled BIC systems in acoustic energy harvesting. In general, this work presents some interesting results and expands the application range of BICs into energy harvesting. The manuscript can be accepted for publication after addressing the following concerns:

1. In the term of acoustic energy harvesting, the authors primarily use open voltage, especially when comparing the performance of different configurations. The authors shall provide more details to justify why we can use open voltage to describe energy harvesting.

We have added an explanation to justify the use of open voltage.

Experimental verification of maximum output voltage

The response of the harvesting system is related to both the acoustic pressure within the resonant structure and the characteristics of the PZT. The impedance of the circuit, and therefore, amongst others, its resistance, determines the system response in terms of output power. To eliminate the effect of resistance on the output signal, we use the open circuit voltage. As the output voltage is linearly correlated with the pressure enhancement, we use it as the physical quantity to characterize and compare the systems presented. Three configurations are manufactured and compared with each other. The first configuration is a 3D printed sound hard termination with mounted PZT to determine the resonance of the mounted PZT. . . .

2. In the Introduction, the authors emphasize the importance of acoustic energy harvesting in the field of energy harvesting. To provide a comprehensive overview of this topic, it is recommended to include the latest developments in acoustic energy harvesting, including output power and open voltage. This will help readers better understand the performance characteristics of acoustic energy harvesting and its practical application potential.

We have included the latest developments in acoustic energy harvesting regarding open voltage and output power.

Introduction

Recent advancements have focused on various mechanisms and materials to enhance the efficiency and practicality of acoustic energy harvesting systems. For instance, piezoelectric and flexoelectric materials, often integrated with phononic crystals and metamaterials, have significantly improved energy conversion efficiency [27-29]. Devices incorporating these materials can harvest energy from low-frequency sounds, which are prevalent in urban environments and typically difficult to absorb or isolate using conventional methods. Due to the low energy density of acoustic waves, acoustic energy enhancement measures are inevitable. Thus, the use of Helmholtz resonators [30, 31] and straight tube resonators [32] has been explored. Moreover, acoustic metamaterials, with their ability to manipulate and focus sound waves in often unprecedented ways, are a natural fit for these applications [21, 33, 34]. For trending self-powered acoustic systems, increased energy density is also inevitable [26, 35-38]. Energy density amplification of several orders of magnitude has been achieved using the latest energy enhancement devices [39, 40]. A further increase in acoustic energy density is needed to make low-density acoustic waves a widely usable energy source, though.

. . .

Regarding the characterization of acoustic energy harvesters, recent work has focused mainly on the following physical quantities: The system's open circuit voltage [56-58], its output power or output power density [26, 59], and its energy conversion efficiency [37 56, 60, 61]. The open-circuit voltage is the highest voltage that the harvester can generate in the absence of an external load. As the open-circuit voltage is

related to the pressure increase of the resonant structure and the properties of the PZT, it is a suitable quantity to characterize pressure-enhancing structures.

Here, we introduce BIC-based acoustic energy harvesting as a technically viable mechanism to address some of the energy requirements of future applications. Theoretical and numerical investigations provide fundamental insights into the formation mechanism of the F-W BIC and the resonant cavity in which it is housed. Furthermore, we analyze the BIC harvester numerically and experimentally regarding output voltage and compare it with a state-of-the-art Helmholtz resonator harvester. Impedance tube measurements and laser Doppler vibrometry help to verify the results. We compare the two harvesting systems on the basis of open-circuit voltage. We then present optimum resistance studies for both harvesters and compare their measured output power. Finally, we present a novel coupling-enhanced BIC harvester, where appropriate experiments highlight its outstanding performance.

3. The authors measure the open-voltage of three different structural configurations to characterize acoustic energy harvesting. However, as voltage does not directly represent energy, using it as an evaluation parameter may not provide an accurate evaluation, energy harvesting should be characterized by parameters such as output power or energy density. Please provide output power of three different configurations.

The output power for the three different configurations has been added. In addition, three resistors of $100\ \Omega$, $1\ \text{k}\Omega$ and $10\ \text{k}\Omega$ are measured and the corresponding output powers are calculated.

After establishing the reference, we measure the FRC (Fig. 3b). The ASD values of the velocity and the output voltage maximize at the same frequency $f = 2068.5\ \text{Hz}$, which means that there is a slight shift of $f = 5.5\ \text{Hz}$ compared to the maximum absorption at $f = 2074\ \text{Hz}$. For a white noise excitation, we obtain a maximum ASD value of the velocity of $\tilde{v} = 0.16\ \text{mm/s}/\sqrt{\text{Hz}}$ and a maximum ASD value of the output voltage of $\tilde{U} = 0.25\ \text{mV}/\sqrt{\text{Hz}}$. As expected, the second local maximum appears at $2329\ \text{Hz}$ with values of $\tilde{v} = 0.046\ \text{mm/s}/\sqrt{\text{Hz}}$ and $\tilde{U} = 0.063\ \text{mV}/\sqrt{\text{Hz}}$, similar to the ones of the PZT resonance observed before. Furthermore, we excite the system with the determined target frequency using a sinusoidal signal at $f = 2068.5\ \text{Hz}$. The corresponding velocity and output voltage are displayed in Fig. 3c for $1.5\ \text{ms}$. Throughout all measurements, the LDV captures the structural dynamics of the PZT. Numerical data indicate that, compared to a Helmholtz resonator peaking at the same resonant frequency, the output voltage is 192 times higher.

Experimental verification of maximum output power.

The output power of the FRC harvesting system is studied numerically to analyze its harvesting characteristics. Therefore, an optimal resistance study is performed. The results are shown in Supplementary Information, Section S 5, Fig. S 7. Figure 3d shows the harvesting characteristics of the FRC system. The system is again excited at $f = 2068.5\ \text{Hz}$, but with a $10\ \Omega$ resistor placed in the circuit. The voltage is measured across the resistor to determine the electrical current. The velocity of the PZT increases to $16.7\ \text{mm/s}$ as the voltage drops to $0.5\ \text{mV}$. In this way, an electric current of $50\ \mu\text{A}$ is measured. More importantly, the electrical power extracted from the circuit is $12.5\ \text{nW}$. Measurements of three additional resistors, $100\ \Omega$, $1\ \text{k}\Omega$, and $10\ \text{k}\Omega$, are provided for better evaluation of the system. This results in the following output powers of $121\ \text{nW}$, $822\ \text{nW}$, and $479\ \text{nW}$ respectively. The maximum output power of $822\ \text{nW}$ with a $1\ \text{k}\Omega$ resistor placed in the circuit is in line with the trend of the numerical prediction. The system's energy conversion efficiency is therefore around 10%. Furthermore, the sound hard termination with mounted PZT has a maximum output power of $50\ \text{zW}$ and the Helmholtz resonator of $87\ \text{nW}$. In summary, the FRC outperforms the Helmholtz resonator by a factor of about 10 in terms of output power.

S 5 Optimum resistance study

We carry out an optimum resistance study to determine which resistance will give the highest output power P_o from the FRC harvester. The output power for different resistances from $10\ \Omega$ to $10\ \text{k}\Omega$ is shown in Fig. S 7.

Fig. S 7. Optimum resistance analysis. FRC harvester output power over a resistance range from 10Ω to $10 \text{ k}\Omega$.

The maximum output power is achieved around 2511Ω . The numerical results are an approximation of the PZT used in the experimental setup, as we can only estimate its material parameters. Thus, the order of $1 \text{ k}\Omega$ is expected for the maximum output power of the experimental setup.

- The authors claim that the measured absorption coefficient in Figure 1d is lower than the numerical value due to additional damping introduced by the PZT. It is recommended to provide absorption coefficient data for the structure without the PZT to verify this explanation.

We have measured the FRC without PZT. The results are similar. Thus, we have corrected the statement.

We explain this behavior by the influence of additional structural damping, which is not represented by the numerical data using sound hard boundary conditions.

- The authors measure the output power of the FRC system as 12.5 nW with an external load resistance of 10Ω . Please clarify the reason for selecting 10Ω as the external load resistance and indicate whether an optimal resistance calculation was conducted.

We have added an optimal resistance calculation and also measured resistances of various magnitudes to verify the numerical predictions. Please refer to the answer to question number 3 above.

- Please provide the energy conversion efficiency of the system, as this is an important parameter for acoustic energy harvesting systems.

We have added the energy conversion efficiency of the FRC system. Please refer to the answer to question number 3 above.

- The results of the numerical simulation indicate that the output voltage of the FRC structure is 192 times higher than that of the HR resonator. It is recommended to include relevant parameters for output power. Additionally, please explain whether losses were introduced in the simulation and how they were set.

We have added the output powers of both systems: In summary, the FRC outperforms the Helmholtz resonator by a factor of about 10 in terms of output power. Please refer to the answer to question number 3 above.

We have used thermo-viscous losses in all simulations. The exact formulation is explained in the following section:

Numerical simulations. All simulations in this article are performed with the commercial software COMSOL Multiphysics (Acoustics Module, Structural Mechanics Module, and AC/DC Module). The speed of sound and the air density is $c_0 = 343$ m/s and $\rho_0 = 1.2$ kg/m³, respectively. We consider the walls of the cavity and the walls of the waveguide to be rigid and thus apply sound hard boundary conditions. We also consider thermo-viscous losses in our system. We apply the no-slip condition for the velocity field and an isothermal condition for the temperature at the walls of the cavity. To ensure non-reflective boundary conditions at the ends of the waveguide, we use perfectly matched layers. An acoustic-structure boundary is applied at the interface between the PZT and the fluid medium. We use strong coupling. This means that the kinetic condition and the kinematic condition at the solid-fluid interface are taken into account. The PZT is modeled using a brass plate facing the fluid medium (density $\rho = 8900$ kg/m³, Young's modulus $E = 100$ GPa, and Poisson's ratio $\nu = 0.3$) with a layer of Lead Zirconate Titanate (PZT-5H: $\rho = 7500$ kg/m³, piezoelectric strain constants $d_{31} = -274$ pC/N and $d_{33} = 593$ pC/N, relative permittivity $\epsilon_{11} = \epsilon_{22} = 1704.4$ and $\epsilon_{33} = 1433.6$) applied to it. In the experimental setup, the PZT is bonded to the printed structure and, assuming the bond is rigid, we apply a fixed constraint to the interface between the structure and the brass plate of the PZT. The Electrostatics interface is used to model the PZT. In addition, an electrical circuit is embedded in the simulation to model the output power of the system in relation to a given resistance.

8. In conclusion, the authors claim that they achieve impedance matching between the medium and the PZT, please explain how this is accomplished. As far as I know, the acoustic impedance of the medium and PZT are both significantly higher than that of air, allowing them to be considered as hard boundaries.

We have clarified our statement.

Abstract

Due to their high-quality factor, they offer exceptional energy enhancement properties, resulting in highly localized energy. As a result, bound states in the continuum are particularly well suited to harvesting energy with a high degree of efficiency.

Discussion

Our studies show that the presented BIC harvester is superior to a conventional Helmholtz resonator harvester by a factor of 2.2 in terms of ASD of the output voltage. The BIC results in an increase in stored energy. Adding a PZT with a resistive load results in a more efficient coupling of energy into the PZT. Thus, we are able to achieve such an enhancement thanks to the improved impedance matching between the medium and the PZT [50]. Furthermore, we demonstrate the harvesting performance of the proposed BIC harvester by connecting a resistor to the terminals of the PZT to simulate an electrical load.

9. Since this work focuses on acoustic BICs, some relevant works on this topic shall be cited. Some examples are listed as follows
 - (a) B. Jia et al., Phys. Rev. Appl. 19, 054001 (2023)
 - (b) Z. Zhou et al., Phys. Rev. Lett. 130, 116101 (2023)
 - (c) A. A. Lyapina et al., Journal of Sound Vibration 421, 48 (2018)

We have added the recommended works and also other relevant ones.

Specific Response to Reviewer #2:

This paper presented theoretical, numerical, and experimental studies on bound state acoustic harvesters. The overall study is very rigorous, but I would like to make the following suggestions here.

1. Please center the image in Fig.2 and Fig.4.

We have centered the image in Fig.2 and Fig.4.

2. In this paper, white noise is used for testing. It is well known that the power spectral density of each frequency component of white noise is constant, and the power spectral density of Brown noise is proportional to the reciprocal frequency. So how would the results be different if Brownian noise was used for the test?.

Preliminary measurements have shown that the system response, i.e. the voltage output, is linear when the amplitude of the excitation signal depends on its frequency. Recognizing this transfer behavior, a different excitation spectrum will result in a different system response but will not provide further insight. Consequently, the velocity and output voltage peaks around 2300 Hz in Figs. 3a and 3b would be reduced.

Specific Response to Reviewer #3:

This article presents a novel energy harvester based on BICs. The paper provides a detailed introduction to the design and experimental process. The research findings are innovative. This BICs energy harvester is inspiring for applications. I have the following unclear content.

1. How does the BICs energy harvester introduced in the paper improve impedance matching between the medium and PZT and enhance acquisition performance? This is not clearly stated in the paper. Suggest adding explanations.

We have clarified our statement.

Abstract

Due to their high-quality factor, they offer exceptional energy enhancement properties, resulting in highly localized energy. As a result, bound states in the continuum are particularly well suited to harvesting energy with a high degree of efficiency.

Diskussion

Our studies show that the presented BIC harvester is superior to a conventional Helmholtz resonator harvester by a factor of 2.2 in terms of ASD of the output voltage. The BIC results in an increase in stored energy. Adding a PZT with a resistive load results in a more efficient coupling of energy into the PZT. Thus, we are able to achieve such an enhancement thanks to the improved impedance matching between the medium and the PZT [50]. Furthermore, we demonstrate the harvesting performance of the proposed BIC harvester by connecting a resistor to the terminals of the PZT to simulate an electrical load.

2. The simulation model used the commercial software COMSOL Multiphysics (Acoustic Module and Structural Mechanics Module). The performance of PZT and BICs is related to the structural connection method and environment. How is this aspect understood and designed? Suggest providing a detailed introduction.

We have added a more detailed explanation in the Methods section.

Numerical simulations. All simulations in this article are performed with the commercial software COMSOL Multiphysics (Acoustics Module, Structural Mechanics Module, and AC/DC Module). The speed of sound and the air density is $c_0 = 343$ m/s and $\rho_0 = 1.2$ kg/m³, respectively. We consider the walls of the cavity and the walls of the waveguide to be rigid and thus apply sound hard boundary conditions. We also consider thermo-viscous losses in our system. We apply the no-slip condition for the velocity field and an isothermal condition for the temperature at the walls of the cavity. To ensure non-reflective boundary conditions at the ends of the waveguide, we use perfectly matched layers. An acoustic-structure boundary is applied at the interface between the PZT and the fluid medium. We use strong coupling. This means that the kinetic condition and the kinematic condition at the solid-fluid interface are taken into account. The PZT is modeled using a brass plate facing the fluid medium (density $\rho = 8900$ kg/m³, Young's modulus $E = 100$ GPa, and Poisson's ratio $\nu = 0.3$) with a layer of Lead Zirconate Titanate (PZT-5H: $\rho = 7500$ kg/m³, piezoelectric strain constants $d_{31} = -274$ pC/N and $d_{33} = 593$ pC/N, relative permittivity $\epsilon_{11} = \epsilon_{22} = 1704.4$ and $\epsilon_{33} = 1433.6$) applied to it. In the experimental setup, the PZT is bonded to the printed structure and, assuming the bond is rigid, we apply a fixed constraint to the interface between the structure and the brass plate of the PZT. The Electrostatics interface is used to model the PZT. In addition, an electrical circuit is embedded in the simulation to model the output power of the system in relation to a given resistance.